# Non-Calcified Coronary Artery Plaque on Coronary Computed Tomography Angiogram: Prevalence and Significance

**Bandar Alyami** [1] , **Matthew Santer** [1] , **Karthik Seetharam** [2], **Dhivya Velu** [2], **Eswar Gadde** [3], **Bansari Patel** [1] **and Yasmin S. Hamirani** [2,*]

1   Department of Medicine, School of Medicine, West Virginia University, Morgantown, WV 26506, USA; banalyami@gmail.com (B.A.); mjsanter@hsc.wvu.edu (M.S.); bansari.patel@hsc.wvu.edu (B.P.)
2   Department of Cardiology, Heart and Vascular Institute, West Virginia University, Morgantown, WV 26506, USA; skarthik87@yahoo.com (K.S.); dhivya.kuzhandaivelu@hsc.wvu.edu (D.V.)
3   Department of Medicine, West Virginia University, Charleston, WV 25304, USA; eswar.gadde@hsc.wvu.edu
*   Correspondence: yh961@rwjms.rutgers.edu or yasmin.hamirani@wvumedicine.org; Tel.: +1-(304)-598-4651; Fax: +1-(304)-285-1987

**Abstract:** Objective: We aimed to assess the prevalence of non-calcified plaque (NCP) on computed tomography angiography (CCTA) in symptomatic and asymptomatic individuals. In addition, we seek to compare plaque assessment on CCTA with intravascular ultrasound–virtual histology (IVUS-VH) and to assess the prognostic value of non-calcified plaques (NCPs). Background: The CCTA can characterize coronary plaques and help quantify burden. Furthermore, it can provide additional prognostic information which can enable further risk stratification of patients. Methods: We performed a broad comprehensive review of the current literature pertaining to CCTA and primarily isolated NCP in symptomatic and asymptomatic patients. In addition, our review included studies correlating plaque on CT with IVUS-VH. Conclusions: NCP is the initial precursor of calcified plaque and serves as a prominent marker of early coronary atherosclerosis. By detecting NCP during early stages, several measures can be implemented which can alter the evolutionary course of the underlying disease. This can potentially lead to a lower incidence of cardiovascular events.

**Keywords:** chest pain; cardiac CT calcium score; coronary computed tomography; angiography; coronary plaques; systematic review





## 1. Introduction

Coronary artery disease is widely established as a leading cause of mortality in developed countries [1]. Assessment of calcium on non-contrast CT scans is a well-recognized traditional approach for predicting cardiovascular (CV) outcomes in asymptomatic individuals [2]. Interestingly, a number of new technological advancements and innovations in non-invasive imaging can further delineate patients with high risk for future major cardiovascular events (MACE). One of these promising advances is the ability to distinguish coronary artery plaques through non-invasive approaches, which include non-invasive coronary CT angiography (CCTA) (Figure 1).

The most recent ACC/AHA guidelines for chest pain assessment have mentioned the importance of coronary CTA as one of the first-line tests to evaluate obstructive coronary artery stenosis [3,4]. Regardless of the magnitude of coronary artery stenosis identified by CCTA, the overall plaque burden has been linked to poor prognosis in multiple prior reports [5–8]. In addition, SCCT guidelines on CCTA interpretation recommend visual plaque quantification in the CAD RADS reporting scheme [9]. High-risk plaque (HRP) on CCTA includes positive remodeling (PR), napkin ring lesions, spotty calcification, and low-attenuation non-calcified plaque (LAP) with a HU < 30 [10–12] (Figure 2).

With prominent advances in plaque quantification and characterization on CCTA using artificial intelligence/machine learning-based algorithms or pipelines, the inter-

observe, intra-observer, and inter-scanner variations in reporting plaque quantification can be minimized. This will facilitate more population-based research in symptomatic and asymptomatic individuals to understand the importance of the presence, distribution, and characteristics of plaque on coronary CTA in predicting future cardiovascular outcomes.

In this review, we aim to review the current literature on non-calcified plaque (NCP), including LAP, in CCTA and its overall impact on CV outcomes.

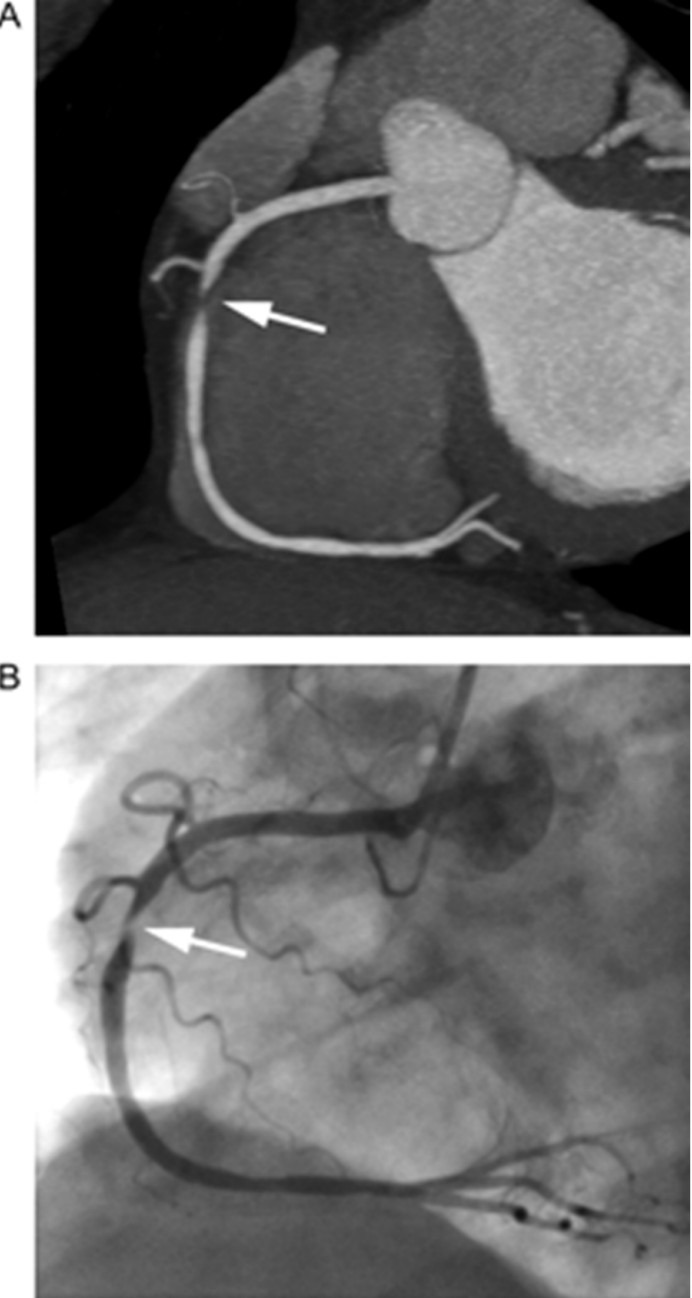

**Figure 1.** Visualization of coronary artery luminal stenosis by contrast-enhanced computed tomography angiography. Here, a high-grade stenosis of the right coronary artery is present. (**A**) Coronary computed tomography angiography (arrow = high-grade stenosis of the right coronary artery). (**B**) Invasive coronary angiogram of the right coronary artery (arrow = stenosis). Source: Achenbach et al. Imaging of coronary atherosclerosis by computed tomography, *Eur Heart J*, Volume 31, Issue 12, June 2010, Pages 1442–1448 by permission of Oxford University Press [13].

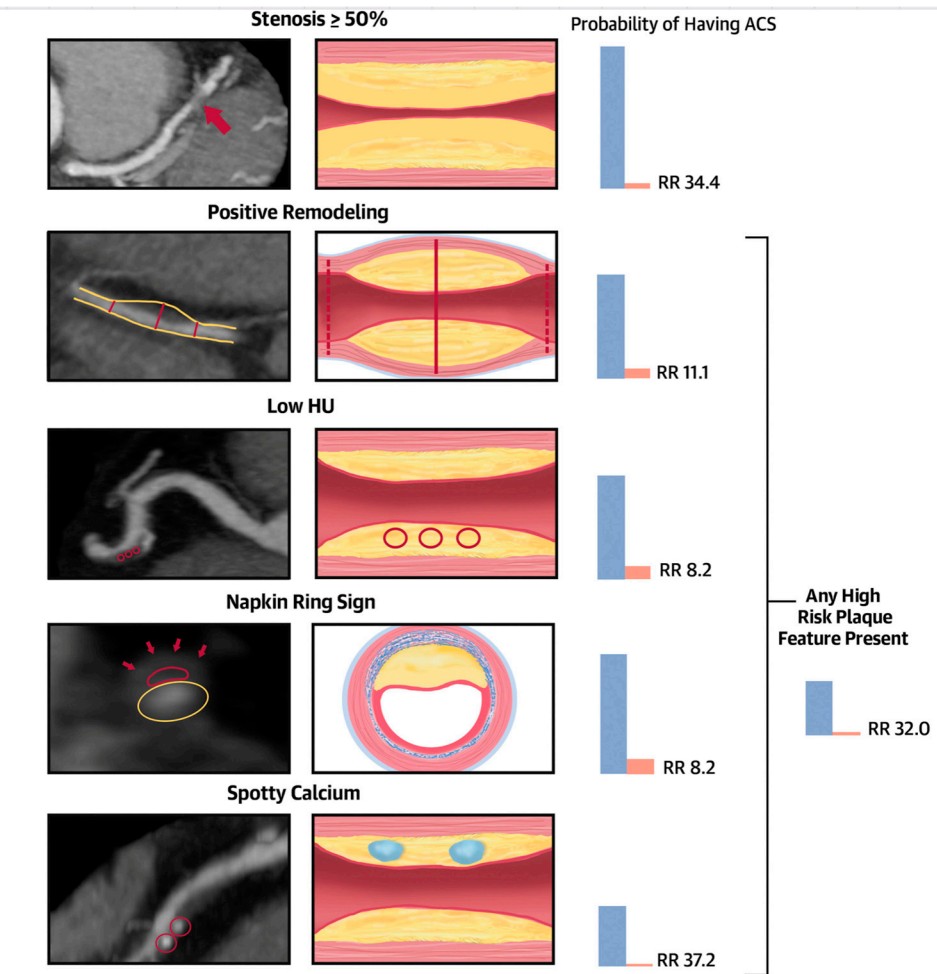

**Figure 2.** Central illustration. Significant stenosis and high-risk coronary plaque features and their association with the probability of ACS during index hospitalization. Stenosis ≥ 50%: severe stenosis of the mid-left anterior descending coronary artery (red arrow). Positive remodeling: noncalcified plaque with positive remodeling in the distal right coronary artery. The 2 dotted red lines demonstrate the vessel diameters at the proximal and distal references (both 1.8 mm), and the solid red line demonstrates the maximal vessel diameter in the mid portion of the plaque (2.7 mm). The remodeling index is 1.5. Low Hounsfield units (HU) plaque: partially calcified plaque in the mid-right coronary artery with low <30 HU plaque. The red circles demonstrate the 3 regions of interest, with mean computed tomography (CT) numbers of 22 HU, 19 HU, and 20 HU. Napkin-ring sign: napkin-ring sign plaque in the mid-left anterior descending coronary artery. Schematic cross-sectional view of the napkin-ring sign. The red line demonstrates the central low HU area of the plaque adjacent to the lumen (yellow ellipse) surrounded by a peripheral rim of the higher CT attenuation (red arrows). Spotty calcium: partially calcified plaque in the mid-right coronary artery with spotty calcification (diameter < 3 mm in all directions; red circles). Patients with significant CAD showing ≥50% stenosis were at a significantly higher risk of experiencing an ACS during their hospitalization, being 34 times more likely compared to those without significant CAD. Furthermore, the presence of high-risk plaque, including positive remodeling (RR 11.1), low Hounsfield unit (HU) or napkin-ring sign (NRS) features (both with a RR of 8.2), spotty calcium (RR 37.2), or at least 1 high-risk plaque feature (RR 32), was associated with an elevated relative risk of ACS. ACS = acute coronary syndromes; RR = relative risk. Source: Puchner SB, High-Risk Plaque Detected on Coronary CT Angiography Predicts Acute Coronary Syndromes Independent of Significant Stenosis in Acute Chest Pain: Results From the ROMICAT-II Trial, Volume 64, Issue 7, 19 August 2014, Pages 684–692 [11].

## 2. Materials and Methods

We performed a broad comprehensive review of the current literature pertaining to CCTA and primarily isolated NCP in symptomatic and non-symptomatic patients. In addition, our review included studies correlating plaque on CT with IVUS-VH.

The selection of studies for this systematic review includes initial searches that were conducted on the PubMed database, utilizing a combination of relevant Medical Subject Headings (MeSH) terms and keywords such as "non-calcified coronary artery plaque", "coronary CT angiogram", "prevalence", and "significance". The search aimed to identify studies published in the English language without imposing limitations on publication dates.

Inclusion criteria were defined to encompass studies specifically investigating non-calcified coronary artery plaques as detected by CTA and providing insights into both their prevalence and clinical relevance. Studies falling under the categories of reviews, case reports, conference abstracts, and those lacking essential data were excluded. To uphold the credibility of the review, only peer-reviewed, full-text articles were incorporated.

## 3. Results

### 3.1. Non-Calcified Plaque (NCP) Detected by CCTA in Symptomatic Patients

Hausleiter et al. analyzed 161 patients and noncalcified coronary plaques were found in 48 (29.8%). Among these, 38 (23.6%) also had coronary calcifications, while noncalcified plaques as the sole manifestation of CAD were observed in 10 (6.2%) patients. Patients with noncalcified plaques exhibited higher total cholesterol, LDL, and CRP levels, along with a trend toward increased diabetes mellitus. Most noncalcified plaques resulted in lumen narrowing of less than 50%. In the remaining 113 patients, CAD was ruled out in 53 (32.9%), while 60 (37.3%) had calcifications without noncalcified plaque [14]. Nance Jr et al. evaluated 458 patients at low-to-intermediate risk for CAD presenting with chest pain. During a 13-month median follow up, the presence of plaque was associated with higher adverse outcomes, the Hazard ratios (HRs) of mixed vs. NC vs. calcified plaque were 86.96 vs. 58.06 vs. 32.94; all with $p$-values < 0.05, respectively [15]. In their population, patients with isolated NCP had a significantly higher likelihood of adverse cardiac events of 5% (hazard ratio 151.77, $p < 0.01$).

Liu et al. performed a subgroup analysis on 260 individuals from the ROMICAT II trial and analyzed segmental coronary plaque analysis on CTA. Among 888 segments with plaque, 391 segments (44%) showed the presence of HRP. Segments associated with HRP had higher total plaque volume, LAP volume, total plaque burden, and higher remodeling index [16]. The odds ratio (OR) associated with LAP was 1.12 (95% CI 1.04–1.21). Al-Muhaidb et al., in a retrospective study, explored the prevalence of NCP in 299 patients with a coronary artery calcium score of zero (CACS). The prevalence of NCP was found to be 6.4%. A strong association of NCP with male sex, HTN, and smoking (all $p$ values < 0.005) was seen [17].

In a post hoc analysis of 1769 patients with stable chest pain from the SCOT-HEART cohort, Williams et al. examined the presence of adverse plaque (positive remodeling and LAP) and its association with outcomes at 5 years. Unsurprisingly, the primary events encompassing fatal and non-fatal myocardial infarction (MI) were higher in the presence of LAP compared to other plaque types (HR: 1.6, 95% CI: 1.1–2.34, $p = 0.014$). Also, more events were noted in patients with adverse plaque despite having CACS < 100 AU with a HR of 3.38 (CI 1.13 to 10.08, $p = 0.03$). The 41 patients with fatal or nonfatal MI at follow-up had significantly higher LAP burden of 7.5% [4.8–9.2] versus 4.1% [0–6.8] in patients without events (HR of 1.6; 95% CI = 1.1–2.3, $p < 0.001$). The overall presence of LAP burden of >4% increased the likelihood of MACE by five times in this population [18]. In another analysis on the same cohort by Osborne-Grinter et al., CACS of zero was seen in 36% of patients. Overall, 4% of the patients with zero CACS had nonobstructive CAD while 2% had obstructive CAD. In addition, 2% had LAP, and 13% had a LAP burden > 4%. Increases in NCP and LAP burden were found in no or low CACS subsets but not in

medium-high CACS groups. At follow-up, 10% of the total reported MIs (41 patients) were solely found in the zero-CACS group [19]. Building on a similar clinical question, Villines et al. evaluated 10,037 symptomatic patients without CAD from the CONFIRM registry and identified 51% patients with a <u>CACS</u> of 0. A large percentage (84%) had no underlying CAD while the remaining 13% were diagnosed with nonobstructive NCP. Amongst the NCP group, 3.5% had luminal stenosis > 50% and 1.4% had luminal stenosis > 70%. In patients diagnosed with non-calcified plaque (NCP), the study found that when more than 50% stenosis was identified, which occurred in 3.9% of cases, the risk of adverse events was significantly higher, with a hazard ratio of 5.7 (confidence interval: 2.5–13.1, $p < 0.001$). Furthermore, an analysis using the receiver–operator characteristic (ROC) curve demonstrated that there was no significant additional prognostic value in incorporating the coronary artery calcium (CAC) score when compared to assessing the extent of NCP on coronary computed tomography angiography (CCTA) for composite endpoints [20] (Table 1).

**Table 1.** Studies involving assessment of non-calcified plaque and their association with cardiovascular outcomes in symptomatic patients.

| Reference | Publication (Month, Year) | Study Design | No. of Patients | Age (Years) | Male (%) | Patient Population | Prevalence of NCP | Follow-Up Available Y/N If Y, Duration: | Key Findings of the Study: |
|---|---|---|---|---|---|---|---|---|---|
| Hausleiter et al. [14] | July 2006 | Prospective | 161 | 41–69 | 69.5 | Symptomatic patients at intermediate risk for CAD | 29.8% (48 pts) isolated NCP in 10 (6.2%) | N | -The NCP were the only manifestation of CAD in 6.2% of the study population<br>-Patients with noncalcified plaques were characterized by significantly higher LDL, C-reactive protein levels as well as a trend for more diabetes mellitus. |
| Nance. et al. [15] | September 2012 | Prospective | 458 | $55 \pm 11$ | 36 | Acute chest pain patients at low-to-intermediate risk for CAD | Isolated NCP in 215 (47%) | Y (13 months) | Events during follow-up:<br>-None (plaque absent)<br>-11/215 (5%) (isolated NCP)<br>Independent predictor of MACE:<br>-Extent of plaque (HR 151.77, $p < 0.001$)<br>-Presence of mixed plaque (HR, 86.96; $p = 0.002$) |
| Liu et al. [16] | August 2017 | Randomized Controlled (ROMICAT II trial) | 501 (473 with CCTA | $56.1 \pm 7.8$ | 62.7 | Acute chest pain patients presenting to ED without ischemic EKG changes or troponin elevation | -260/473 (54%)<br>-Isolated NCP 197/260 (75.8%)<br>-At least 1-HRP feature in 166 (63.8%) | N | -Spotty calcification: 151 (58.1%)<br>-Positive remodeling: 55 (21.2%),<br>-Low HU plaque: 39 (15.0%),<br>NRS: 26 (10.0%) |
| Al-Muhaidb et al. [17] | May 2021 | Retrospective | 299 with 0 CACS | | | Chest pain with no prior history of CAD | Isolated NCP 6.4% (19/299) | Y (2 years) | -Patients with NCP:<br>52.6% had no stenosis;<br>26.3% had <25% stenosis;<br>21% had 25–50% stenosis;<br>none had >50% stenosis.<br>Strong correlation of NCP was noted with:<br>-Male sex ($p = 0.001$);<br>-Smoking ($p = 0.004$);<br>-Hypertension, ($p = 0.042$). |
| William et al. [18] | 2020 | Randomized Controlled (SCOT-HEART Trial) | 1769 | $58 \pm 10$ | 56 | Patients with stable chest pain | - | Y (4.7 years) | LAP and associations:<br>-CACS (r = 0.62; $p < 0.001$);<br>-CV risk score (r = 0.34; $p < 0.001$);<br>-luminal stenosis (r = 0.83; $p < 0.001$).<br>-LAP burden was the strongest predictor of MI (aHR, 1.60 (95% CI, 1.10–2.34) per doubling; $p = 0.014$), irrespective of CV risk score, CACS, or stenosis percentage.<br>>4% LAP burden → 5 × more likely to have MI (HR, 4.65; 95% CI, 2.06–10.5; $p < 0.001$). |

**Table 1.** *Cont.*

| Reference | Publication (Month, Year) | Study Design | No. of Patients | Age (Years) | Male (%) | Patient Population | Prevalence of NCP | Follow-Up Available Y/N If Y, Duration: | Key Findings of the Study: |
|---|---|---|---|---|---|---|---|---|---|
| Osborne-Grinter et al. [19] | 2022 | Randomized Controlled (SCOT-HEART Trial) | 1769 529 (36%) with 0 CACS | 58 ± 10 | 56 | Patients with stable chest pain | Isolated NCP 14% | Y (5 years) | -14% → non-obstructive CAD. -2% → obstructive CAD. -2% → adverse plaque visually. -13% → LAP burden > 4%. -41 MI in total population,4 in zero CACS (10%). |
| Villines et al. [20] | 2011 | CONFIRM registry | 10,037 5128 (51%) with 0 CACS | 57 ± 12 | 56 | Symptomatic patients without known CAD | Isolated NCP 13% | Y (2.1 years) | Patients with zero CACS: -13% → nonobstructive stenosis -3.5% → >50% stenosis -1.4% → >70% stenosis -3.9% with a CACS zero and ≥50% stenosis experienced an event (HR: 5.7; 95% CI: 2.5 to 13.1; $p < 0.001$) vs. 0.8% of patients with CACS zero and no obstructive CAD |

Abbreviations: NCP (non-calcified plaque), CACS (coronary artery calcium score), HRP (high-risk plaque), MACE (major adverse cardiovascular events), ROC (receiver operating characteristics), aHR (adjusted hazard ratio), LAP (low-attenuation plaque), CAD (coronary artery disease).

### 3.2. Non-Calcified Plaque (NCP) Detected by CCTA in Asymptomatic Patients

Multiple studies have observed NCP prevalence in asymptomatic patients (Table 2). Rodriguez et al. examined 202 patients [21] at low-to-intermediate risk for CAD. Male sex was associated with a higher total plaque index compared to women (42.06 mm$^2$ $\pm$ 9.22 vs. 34.33 mm$^2$ $\pm$ 8.35; $p < 0.001$). Similarly, patients with elevated LDL level ($\beta = 0.04$ mm$^2$/mg/dL; $p = 0.02$), elevated systolic blood pressure ($\beta = 0.80$ mm$^2$/10 mm Hg; $p = 0.03$), and DM ($\beta = 4.47$ mm$^2$; $p = 0.03$) were found to have a significantly higher NCP. Nezarat et al. [22], in patients younger than 40 years old, noted both calcified plaque as well as NCP to be elevated in DM patients as compared to non-diabetics (19% vs. 58%; $p < 0.001$). Despite a CACS of 0, patients with DM had a higher prevalence of NCP (46%, $p < 0.0001$). On quantitative plaque assessment, all volumes in the NCP type were threefold higher in the presence of DM.

Kral et al. evaluated healthy patients with a family history of premature CAD using dual-source CT (DS-CT). A higher total plaque burden was identified in men vs. women (57.8% vs. 35.8%, $p < 0.0001$). The NCP volume constituted most of the total plaque volume in men (>70%) and women (>80%) [23]. Cho et al. evaluated asymptomatic patients with 0 CACs who showed NCP on CCTA. They revealed that age, male gender, DM, hypertension, and dyslipidemia were significantly associated with the presence of NCP (all $p < 0.05$). However, during follow-up of 313 patients with NCP matched with patients with no NCP, no difference in all-cause death or composite outcome of cardiac death, MI, unstable angina requiring hospitalization, and revascularization occurred after 90 days from index CCTA in both group [24]. Lee et al. assessed a cohort of 441 subjects with 0 CACs and identified that approximately 0.2% of patients who had cardiac events during the follow-up period had >1 coronary segment with NCP. There was a significantly lower mean CT HU and higher remodeling index among these patients [25]. Yang et al. conducted a retrospective, long-term follow-up study aimed at assessing the progression of coronary plaque and its influence on cardiac events among asymptomatic individuals with diabetes mellitus (DM). In a cohort of 197 patients, with an average age of 63.1 years and 60% being male, and a median follow-up duration of 41.8 months, it was observed that patients with a CAC score greater than 10 exhibited an increase in the volume of densely calcified coronary calcium. In contrast, patients with a CACS of 10 or lower showed a more significant increase in the volume of low-attenuation "lipid-rich" plaque components between successive coronary computed tomography angiography (CCTA) scans. The study found that the presence of a CACS greater than 10 was an independent predictors of cardiac events. Asymptomatic individuals with DM exhibited progression of plaque, leading to the eventual development of either symptomatic or asymptomatic coronary artery disease (CAD). Furthermore, the extent of plaque volume increase was associated with subsequent cardiac events, and interestingly, the level of coronary calcification appeared to have an inverse relationship with outcomes in asymptomatic diabetic patients [26].

Jin et al. evaluated the characteristics and predictors of subclinical coronary atherosclerosis and cardiac events in 914 asymptomatic young adults. NCP was the most common type of plaque in asymptomatic young adults and was identified in 6.9% of the population. Forty-six subjects (5.3%) had a CACS of 0 and seventeen (42.5%) had CACS > 0. Multivariate analysis revealed a HR of 2.2 for subclinical coronary atherosclerosis and 49.17% of them for NCP [27]. Yoo et al. evaluated the presence of plaque, plaque characteristics, and CACS in 7515 asymptomatic subjects. When comparing individuals with a Coronary Artery Calcium Score (CACS) of 0 to those with a low CACS, it was observed that those with a low CACS had a notably higher prevalence of non-calcified plaque (NCP) (6.9% vs. 31.5%, with a $p$-value less than 0.001). Among the low-CACS group, independent factors predictive of significant NCP included diabetes mellitus (DM), hypertension, and elevated low-density lipoprotein (LDL) levels (all with $p$-values less than 0.05). Over a median follow-up period of 42 months, individuals in the low-CACS group experienced a significantly higher rate of cardiac events when compared to those in the 0 CACS group (2.6% vs. 0.27%, with a $p$-value less than 0.001) [28].

**Table 2.** Studies involving assessment of non-calcified plaque and their association with cardiovascular outcomes in asymptomatic patients.

| Reference | Month/Year | Design | No. of Patients | Patient Population | Age (Years) | Women (%) | Prevalence of NCP | Median Follow Up (Y/N) | Outcome Variable | Key Findings of the Study |
|---|---|---|---|---|---|---|---|---|---|---|
| Rodriguez et al. [21] | June 2015 | Prospective | 202 | Asymptomatic > 55 years old eligible for statin therapy | $65.5 \pm 6.9$ | 36 | - | Y (8 years) | Assessment of coronary plaque burden | Total plaque index: >In men vs. women by 5.01 mm$^2$; $p < 0.03$); >In patients on increased simvastatin doses (by 0.44 mm$^2$/10 mg; $p = 0.02$). NCP index was positively correlated with: -Systolic BP ($\beta$ = 0.80 mm$^2$/10 mm Hg; $p = 0.03$); -Diabetes ($\beta$ = 4.47 mm$^2$; $p = 0.03$); -LDL ($\beta$ = 0.04 mm$^2$/mg/dL; $p = 0.02$). |
| Nezarat et al. [22] | March 2017 | Prospective, case–control | 181 | Asymptomatic -86 DM patients (25–40 years) with ≥5 years DM type II. -95 non-DM age-/gender-matched. | 25–40 | DM: 56 Non-DM: 46 | In DM with zero CACS 46% | N | Extent, severity, and volumes of coronary plaque in DM patients < 40 years of age | -Prevalence of any plaque: 59% (DM); 20% (no DM). -Total plaque scores, segment involvement scores, and quantitative plaque volume increased in DM. |
| Kral et al. [23] | May 2014 | Prospective | 805 | Gene STAR family study (4000 pts). Asymptomatic patients with no prior CAD were included. | $51.1 \pm 10.8$ | 56 | NCP volume most accounted for in all age -In men < 55 (>70%) -In women < 55 (>80%) | N | Assessment of NCP volumes in patients with family history of early-onset CAD. | -NCP volume increased with age ($p < 0.001$). -NCP higher in men than women ($p < 0.001$). -NCP, as a percentage of total plaque, was inversely related to age ($p < 0.01$). |
| Cho et al. [24] | March 2013 | Retrospective | 4491 with 0 CACS | Asymptomatic subjects undergoing CCTA as part of general health evaluation | $48 \pm 8$ | 43 | 7% (313 pts) | Y (22 months) | Prevalence and prognostic valve of NCP | -No clinical events at 90 days regardless of presence of NCP. |
| Lee et al. [25] | June 2013 | Retrospective | 8668 (6531 with 0 CACS | Asymptomatic patients without prior CAD undergoing CCTA as part of general health evaluation | $49.8 \pm 8.9$ | 44 | 6.75% (441 pts) | Y ($26.4 +/- 14.4$ months) | Cardiac events (death, ACS, or subsequent revascularization) | -All cardiac events 0.18% (12 pts) occurred in patients with NCP and with lower HU and higher RI. |
| Yang et al. [26] | February 2019 | Retrospective | 197 | Asymptomatic with DM and suspected CAD with baseline and follow-up CCTA. | $63.1 \pm 17$ | 40 | - | Y (41.8 months) | -Progression of coronary atherosclerotic plaque. -Association of plaque with cardiac outcomes (cardiac death, non-fatal MI, and revascularization). | -Patients with CACS ≤ 10 had a more pronounced increase in the volume of LAP on CCTA; while -Presence of CACS > 10 had an increase in dense coronary calcium; -10.2% (20 patients) with events (CAC, CAC density and lipid volume independently predicted events). |
| Yoo et al. [28] | December 2011 | Retrospective | 7515 -6040 (80.4%) with 0 CACS -707 (9.4%) with low CACS | | | 30.2 | 0 CACS: 6.9% Low CACS:31.5% | Y (4 years) | Significance of non-calcified coronary plaque. | -Cardiac events in low CACS 2.6% vs. 0.27% in 0 CACS ($p < 0.001$). |

**Table 2.** *Cont.*

| Reference | Month/Year | Design | No. of Patients | Patient Population | Age (Years) | Women (%) | Prevalence of NCP | Median Follow Up (Y/N) | Outcome Variable | Key Findings of the Study |
|---|---|---|---|---|---|---|---|---|---|---|
| Cho et al. [29] | March 2018 | Prospective multicentered registry (CONFIRM long-term study) | 1226 selected from 17,181 pts. | Asymptomatic with no prior CAD history and no intervention < 90 days from CCTA. | 58 ± 12 | 34 | - | Y (5.9 ± 1.2 years) | -Comprehensive CAD assessment by CCTA improves risk prediction for future mortality over a traditional RF model and also when CACS was considered. | -78 deaths at follow-up. -Compared with the traditional RF alone (C-statistic 0.64), CCTA detection of plaque improved incremental prognostic utility beyond traditional RF alone (C-statistics range 0.71–0.73, all $p < 0.05$; incremental $\chi^2$ range 20.7–25.5, all $p < 0.001$). -NCP or mixed plaque in a single segment (HR 2.34, 95% CI 1.23–4.48; $p = 0.010$) or multi-segments (HR 2.50, 95% CI 1.48–4.21; $p = 0.001$) were shown to increase the risk of all-cause death as compared with individuals without any plaque, even after adjustment of traditional RF. |
| Lee et al. [30] | September 2010 | Prospective | 4320 | Asymptomatic individuals who underwent CCA during a routine health check. | 50 ± 9 years | 39 | Prevalence of isolated NCP 5% (801 pts) | N | Determine the prevalence and characteristics of subclinical CAD using CCTA | -Coronary artery plaques were present in 1053 (24%) individuals. -25% (10 pts) with NCP had significant stenosis; most of them were classified into low- or moderate-risk groups according to NCEP risk stratification guidelines. -Amongst men (≤55 years) and women (≤65 years), 30% of subjects with significant stenosis were classified into a low-risk group by NCEP amongst which 60% had low (0 to 100) calcium scores. |
| Nasir et al. [31] | September 2022 | Prospective | 2359 | Asymptomatic individuals from Greater Miami Area. | Mean age 53 years | 50 | Prevalence of isolated NCP 16% | -N | Assess the burden of total coronary plaque, plaque subtypes, and HRP features. | -49% had plaque on CCTA. -58% participants had CACS of 0. -0.8% with CACS 0 had ≥ 50% stenosis; 0.1% had stenosis ≥ 70%. -2.3% of the plaques in 0 CACS were HRP. -Male sex, overweight, and obesity were independent predictors of plaque if CAC was 0. |
| Iwasaki et al. [32] | August 2010 | Retrospective | -502 -224 patients with 0/mild CACS. | Asymptomatic individuals evaluated in an outpatient primary prevention program. | 62.4 ± 10.4 (no CAC) 67.4 ± 8.5 (mild CAC) | 41 | -Prevalence of NCP was 11.1% in patients with no CAC -Prevalence of NCP was 23.4% in mild CAC group ($p = 0.0142$) | -N | Assess prevalence of NCP | -Patients with no CAC were younger. -Multiple plaques were detected in 2.6% of the group with no CAC and 3.7% of the group with mild CAC ($p = 0.5934$). |

Abbreviation: NCP (non-calcified plaque), RI (remodeling index), MI (myocardial infarction), CACS (coronary artery calcium score), CCTA (coronary CT angiography), LAP (low-attenuation plaque), HR (hazard ratio), RF (risk factors), CVD (cardiovascular disease), HU (Housefield unit); RI (remodeling index); DM (diabetes mellitus); CAD (coronary artery disease); NCEP (National Cholesterol Education Program).

Cho and colleagues conducted a study involving 1226 asymptomatic individuals as part of the prospective multicenter international CONFIRM trial. Their aim was to assess how the findings from CCTA could add prognostic value beyond a basic model that already included a set of traditional risk factors (RFs) in combination with the CAC score for predicting long-term all-cause mortality. The study revealed that the presence of non-calcified plaque (NCP) or mixed plaque within a single segment or multiple segments significantly increased the risk of all-cause death when compared to individuals without any plaque. Importantly, this increased risk persisted even after adjusting for traditional risk factors [29]. Lee et al. had CCTA performed in 4320 asymptomatic individuals aged $50 \pm 9$ years as a routine general health check for evaluation of subclinical atherosclerosis. Coronary artery plaques were present in 1053 (24%) individuals and CCTA revealed NCP in 5% of subjects with a CACS of 0 (*n* = 801). Although 25% (*n* = 10) of those with NCP had significant plaque volume, most of them (90%) were further categorized into low- or moderate-risk groups according to National Cholesterol Education Program risk stratification guidelines [30].

Nasir et al., in a recent Miami heart study, identified 0 CACS in 58% of their respective patient population. The prevalence of a CACS of 0 was 29% among participants aged > 50 years and CCTA identified coronary plaque in 16% of CACS 0 patients. Among participants with a CAC of 0, the prevalence of stenosis $\geq$ 50% was only 0.8%, and only 0.1% had stenosis $\geq$ 70% [31]. Iwasaki et al., in their asymptomatic cohort of 505 patients, detected NCP in 11.1% with no CAC vs. 23.4% in the mild CAC group (*p* = 0.0142). Patients with no CAC tended to be younger and had reduced occurrence of DM. Laboratory and medication parameters did not significantly differ between either group. Multiple plaques were detected in 2.6% of the no-CACS group and 3.7% in the group with mild CACS (*p* = 0.5934) [32] (Table 2).

### 3.3. Comparison of CCTA Non-Calcified Plaque (NCP) with IVUS-VH

We reviewed the contemporary literature regarding NCP assessment on CCTA, and its comparison with intravascular ultrasound–virtual histology (IVUS-VH) (Table 3). Obaid et al. assessed the accuracy of CCTA in evaluating plaque components in 57 patients with histologic correlation from 8 postmortem coronary arteries. CT contrast attenuation plaque maps were created in 108 plaques and these maps correlated remarkably with IVUS in regard to plaque composition. In addition, a strong correlation was noted between the two modalities in regard to calculation of necrotic core and total plaque volume. The diagnostic accuracy of CCTA to IVUS-VH was 83% versus 92% for calcified plaque, 80% versus 65% for necrotic core, and 80% versus 79% for fibroatheroma [33]. The CCTA accuracy was similar to IVUS in evaluating plaque fibroatheroma and had superior detection for the fibrotic core (80% vs. 79% and 80% vs. 65%), respectively. However, the IVUS has a better ability to detect calcified plaque and thin-cap fibroatheroma (TCFA) [34].

Carrascosa et al. evaluated 40 patients with CCTA, and compared to IVUS, which was heralded as the gold standard for accuracy evaluation, the total number of plaques was 276. About 99% of them were classified by the CCTA precisely as calcified plaque versus NCP, while 82% of the NCP was subdivided accurately into fibrous and soft plaque [35] using a cut-off value of 88 HU on CCTA. Schepis et al. evaluated the reproducibility of NCP volume measurement by DSCT in comparison to IVUS. This study included 70 patients with 100 individual NCPs (1 to 3 plaques per patient). The difference in the NCP volume by CT measurement between different observers and the variation in the same observer reading was reported to be $6 \pm 5\%$ and $11 \pm 7\%$, respectively. The CT mean plaque volume $89 \pm 66$ mm was comparable to IVUS $90 \pm 73$ mm with a mean difference between both modalities of $1 \pm 34$ mm. Bland–Altman agreement between the two modalities was modest, i.e., $-67$ to $+65$ mm [33]. Hara et al. noted, in 33 patients with 56 proximal plaques, a strong correlation of vessel and lumen size between the modalities ($R^2$ = 0.614, 0.750, respectively) [36]. Furthermore, there was a strong correlation between the percentage of plaque area assessed by MDCT and IVUS ($R^2$ = 0.824).

**Table 3.** Studies comparing the assessment of non-calcified plaque (NCP) with coronary CT angiography (CCTA) and intravascular ultrasound–virtual histology (IVUS-NH).

| Reference | Month/Year | Study Population | Purpose of the Study | Key Findings of the Study | Limitations of CCTA: |
|---|---|---|---|---|---|
| Obaid et al. [34] | August 2013 | 57 | Compare CT generated plaque maps with IVUS-VH. | Correlation between CT and IVUS:<br>Necrotic core: r = 0.41 ($p$ = 0.002);<br>Fibrous plaque: r = 0.54 ($p$ < 0.001);<br>Calcified plaque: r = 0.59 ($p$ < 0.001);<br>Total plaque: r = 0.62 ($p$ < 0.001).<br>Diagnostic accuracy of CT vs. IVUS-VH:<br>Calcified plaque (83% versus 92%);<br>Necrotic core (80% versus 65%);<br>Fibroatheroma (80% versus 79%). | -VH-IVUS could identify TCFA with a diagnostic accuracy between 74% and 82% (depending on the TCFA definition used).<br>-Spatial resolution of CCTA prevents direct identification of TCFA. |
| Carrascosa et al. [35] | March 2006 | 40<br>(Mean age: 52, 80% Males) | Compare plaque composition between DSCT and IVUS. | 276 plaques examined by IVUS and DSCT.<br>-Calcified plaque (using CT cut off of 185 HU identified 273/276 plaques (99%).<br>-Fibrous/soft plaques (using CT cut off of 88 HU identified 192/233 (82%)) | -Results were obtained utilizing a 4-detector scanner with no multicycle reconstruction capability. |
| Schepis et al. [33] | April 2010 | 70<br>-100 individual NCP (1 to 3 plaques per patient) | Compare NCP volumes on DSCT vs. IVUS. | -Mean total plaque volume by DSCT was 89 $\pm$ 66 mm$^3$ (range 14–400 mm$^3$).<br>-Mean total plaque volume by IVUS was 90 $\pm$ 73 mm$^3$ (range 16–409 mm$^3$).<br>The mean difference between DSCT and IVUS was 1 $\pm$ 34 mm$^3$ (range −131–85 mm$^3$).<br>-Correlation between two modalities (r = 0.89, $p$ < 0.001) | -Modest agreement between DSCT and IVUS (Bland–Altman limits of agreement −67 to +65 mm$^3$). |
| Hara et al. [36] | June 2007 | 33 | Accuracy of non-stenotic atherosclerotic assessment using CCTA vs. IVUS. | -56 proximal lesions from 33 patients assessed.<br>-vessel size R$^2$ = 0.614.<br>-lumen size R$^2$ = 0.750.<br>-percentage plaque R$^2$ = 0.824. | - |
| Sakakura et al. [37] | March 2006 | 16 | Plaque characterization in patients within 7 days from ACS using combined CCTA and IVUS | -23 plaques identified by IVUS (6 soft, 11 intermediate, and 6 calcified plaques).<br>-CT HU for these plaques:<br>Soft → 50.6 $\pm$ 14.8 HU<br>Intermediate → 131 $\pm$ 1.0 HU<br>Calcified → 721 $\pm$ 231 HU | - |
| Hur et al. [38] | March 2009 | 39 | Quantification and characterization of obstructive coronary plaque using 64-slice CCTA compared to IVUS | Correlation coefficients:<br>-Lumen r = 0.712<br>-Vessel r = 0.654<br>-Plaque area r = 0.753<br>-Percentage luminal obstruction r = 0.799<br>Mean CT density values for plaque:<br>Soft (n = 10) 54 $\pm$ 13 HU<br>Fibrous (n = 11) 82 $\pm$ 17 HU<br>Mixed (n = 31) 162 $\pm$ 57 HU<br>Calcified plaques (n = 9) 392 $\pm$ 155 HU | CT density measurements not significantly different between soft and fibrous plaques ($p$ = 0.224).<br>-Reliable classification of NCP as vulnerable or stable plaque based on CT density measurements is currently limited. |

**Table 3.** *Cont.*

| Reference | Month/Year | Study Population | Purpose of the Study | Key Findings of the Study | Limitations of CCTA: |
|---|---|---|---|---|---|
| Yang et al. [39] | October 2010 | 46 | Assessment of diagnostic accuracy of DSCT compared to IVUS | -<u>Correlation coefficients:</u><br>-Luminal cross-sectional area 0.82 ($p < 0.01$, CI 0.67–0.89).<br>-External elastic membrane cross-sectional area 0.78 ($p < 0.01$, CI 0.67–0.86).<br>-No significant difference in the distributive characteristics of the lesions in patients with NSTEMI and stable angina pectoris patients was noted. | - |

Abbreviations: IVUS (intravascular ultrasound), NCP (non-calcified plaque), VH (virtual histology), DSCT (dual-source CT scan), STEMI (ST elevation myocardial infarction), NSTEMI (non-ST elevation myocardial infarction), TCFA (thin-cap fibro-atheroma), DSCT (dual-source computed tomography), HU (Hounsfield units).

In a retrospective comparative study, Sakakura et al., in 16 patients, sub-divided 23 plaques into three main types based on the degree of plaque echogenicity. Overall, 6 were soft, 11 intermediate, and 6 calcified with a mean HU of 50.6 ± 14.8 HU, 131 ± 21.0 HU, and 721 ± 231 HU, respectively [37]. Hur et al. used CCTA to measure the tissues' mean plaque density volume in 39 patients for soft (54 ± 13 HU), fibrous (82 ± 17 HU), mixed (162 ± 57 HU), and calcified plaques (392 ± 155 HU). There was no statistical difference in HU between soft and fibrous plaque ($p > 0.05$) [38]. In another study on 46 patients evaluated with both DSCT and IVUS for plaque characterization, the sensitivity, specificity, PPV, and NPV of DSCT compared with QCA were 100%, 98%, 92%, and 100%, respectively. The similar values for DSCT compared with IVUS were 100%, 99%, 95%, and 100%, respectively. These results reflect the ability of DSCT to predict plaque characteristics more accurately. Patients with unstable angina pectoris or ST elevation myocardial infarction primarily were noted to have discrete soft plaques [39] (Table 3).

## 4. Discussion

There is an emerging interest in the assessment of plaques identified by CCTA in both symptomatic and asymptomatic patients. CCTA is being increasingly utilized after subsequent incorporation into the new ACC/AHA guidelines as one of the premier first-line tests in symptomatic patients alongside other traditional stress-testing modalities [3]. Another key reason is the increased availability of artificial intelligence (AI) pipelines based on CCTA plaque algorithms that are now being validated and integrated in the clinical setting. However, CCTA is currently not recommended by both the ACC/AHA and European guidelines in asymptomatic patients [3].

Coronary artery calcium score is performed on non-contrast CT images in asymptomatic patients. However, contrast-enhanced CT scans of coronary arteries with their increased spatial resolution allows plaque assessment regardless of type. Non-calcified plaques (NCPs) are considered precursor lesions for calcified plaque. If attempts are made to identify NCP regression at any early age, a reduction in prevalence of CAD can be achieved along with accompanying reductions in morbidity and mortality [33,36]. After a review of multiple reports, we noted that the absence of coronary calcifications and the prevalence of NCP in symptomatic patients can vary between 6.4 and 44% [14–20], and between 7 and 46% [22–32,34,35] in patients without angina or anginal equivalent symptoms. The numbers vary depending on the population and prevalence of certain risk factors, which include male gender, DM, smoking, and presence of obesity. Another important observation is the rising prevalence of NCP with increased coronary artery calcifications.

Although high-risk plaque features have been defined on CCTA, and plaque characterization has been compared with IVUS-VH on autopsied coronary arteries, there is variation in plaque burden assessment noted on CT with some degree of inter-vendor and inter- and intra-observer variations. This is more a significant issue when the vessel wall is counted as plaque in patients with isolated NCP. However, with augmented AI based plaque analysis, significant improvement is anticipated. Of note, despite the high prevalence of NCP, a reduced occurrence of obstructive CAD is being identified in patients with isolated NCP as compared to calcified plaque [21,32]. On another note, few studies have examined outcomes in regard to MACE in these patients with soft plaque. A limitation of CT is the inability to identify thin-cap fibroatheroma due to limits with resolution. Thus, reliable classification of NCP into vulnerable and stable plaque based on CT density assessment cannot be performed. Despite these concerns, CCTA has a bright future in regard to its potential in risk stratification of individuals with zero CACS and prediction of cardiac events based on plaque characterization. Population-based CCTA studies like MESA and the recent Miami heart study have significantly expanded our understanding of plaque prevalence and its relationship with obstructive CAD. As this population continues to be examined and explored, our knowledge regarding plaque constituency or relationships with ethnicity, familial/genetic tendencies, and risk factor profiles will improve in years to come. One of the ongoing studies addressing this question is the CONFIRM 2 trial.

Plaque that is not calcified on the CT scans can be regressed with treatment and confirmed with follow-up imaging. The clinical utility of this strategy has yet to be established. In a meta-analysis by Andelius et al., patients were subdivided into three cohorts which encompassed intense statin therapy, moderate statin therapy, and a control group. The intensive statin group had a higher reduction in the total plaque volume as well as mean volume. The control group, on the other hand, showed significant increases in total plaque as well as mean plaque volume [39]. Studies should further individualized treatment responses and investigate understudied aspects of anti-inflammatory therapies which could enhance the future treatment approaches of coronary plaque. Longer follow-ups on large populations with a possible time frame of 20–30 years of follow-up is needed to uncover the real significance of identifying NCP in patients, particularly when they are asymptomatic.

## 5. Conclusions

CAD is associated with increased mortality and morbidity and remains a great healthcare concern. Early detection of coronary artery plaque before its progression to calcification can allow early interventions towards regression. This will eventually translate into a lower incidence of future CV events. Dedicated and focused outcome-based research is highly indicated in this area to assess if there is utility of CCTA in symptomatic and asymptomatic patients with zero or minimum CACS. CCTA can be performed with minimal possible radiation exposure.

**Author Contributions:** All authors contributed equally to identification and review of manuscripts pertinent to this review article, its write-up, and editing. All authors have read and agreed to the published version of the manuscript.

**Funding:** This research received no external funding.

**Institutional Review Board Statement:** Not applicable.

**Informed Consent Statement:** Not applicable.

**Data Availability Statement:** Not applicable.

**Conflicts of Interest:** The authors declare no conflict of interest.

## Abbreviations

| | |
|---|---|
| NCP | Non-calcified plaque |
| CACS | Coronary artery calcium score |
| HRP | High-risk plaque |
| MACE | Major adverse cardiovascular events |
| RI | Remodeling index |
| MI | Myocardial infarction |
| HR | Hazard ratio |
| RF | Risk factors |
| CVD | Cardiovascular disease |
| IVUS | Intravascular ultrasound |
| VH | Virtual histology |
| DSCT | Dual-source CT scan |
| STEMI | ST-elevation myocardial infarction |
| NSTEMI | Non-ST elevation myocardial infarction |
| TCFA | Thin-cap fibro-atheroma |
| T2DM | Type II diabetes mellitus |

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
