# Peer review of "Non-Calcified Coronary Artery Plaque on Coronary Computed Tomography Angiogram: Prevalence and Significance"

_tomography, doi:10.3390/tomography9050140_

Round 1

Reviewer 1 Report

One significant area that requires major revisions in the manuscript on the prevalence and significance of non-calcified coronary artery plaque on Coronary CT Angiogram is the lack of thorough analysis and utilization of tables and figures. Incorporating this essential visual representation of data will significantly enhance the manuscript's clarity, comprehensibility, and overall impact.

The manuscript would greatly benefit from the inclusion of well-designed tables and figures to present the key findings and relevant statistical information. Tables can be used to summarize and compare essential characteristics or outcomes across different studies or patient populations. Figures, such as graphs or diagrams, can visually illustrate trends, distributions, or relationships within the data. The strategic use of tables and figures will aid readers in quickly grasping the main findings and facilitate comparisons between different research studies or data sets.

By addressing the lack of table and figure analysis, the revised manuscript will provide a more comprehensive and accessible presentation of the prevalence and significance of non-calcified coronary artery plaque on Coronary CT Angiogram. The inclusion of well-designed tables and figures, along with their thorough analysis and interpretation, will enhance the manuscript's scientific rigor, visual impact, and overall contribution to the field of cardiovascular medicine.

Author Response

Thank you for your valuable input, we have worked on integrating relevant tables and figures into the revised manuscript. We have conducted a thorough review of the tables and figures to provide comprehensive insights into our research. We believe that by addressing this aspect and incorporating these visual aids, our manuscript's scientific rigor and accessibility will be significantly improved.

Once again, we appreciate your thoughtful feedback, as it has played a crucial role in strengthening the quality and impact of our research. 

Reviewer 2 Report

Alyami et al. present a review article about the assessment of the prevalence of non-calcified plaque on computed tomography angiography in symptomatic and asymptomatic individuals. Furthermore, they assessed the prognostic value of NCP. Although the topic is interesting and the manuscript well-written, some considerations need to be clarified.

1. Introduction: the authors cite the ACC/AHA guidelines and SCCT guidelines. Please, also cite the point of view of ESC Guidelines (Knuuti J, et al. ESC Scientific Document Group. 2019 ESC Guidelines for the diagnosis and management of chronic coronary syndromes. Eur Heart J. 2020 Jan 14;41(3):407-477. doi: 10.1093/eurheartj/ehz425).

2. Discussion: Please, suggest the therapeutic strategy that can be useful for the plaque regression based on the current literature.

3. Please, insert a new paragraph “gaps in evidence and future directions” after the discussion.

The authors satisfactorily answered their main question to evaluate the prevalence of non-calcified plaque on computed tomography angiography in symptomatic and asymptomatic individuals and its prognostic value. The topic is original and adds interesting information on importance of detecting the non-calcified plaque during early stages to reduce the future incidence of adverse cardiovascular events.
The conclusions are consistent with the current evidence but the references should be improved.
A figure about high-risk plaque on computed tomography angiography could be added by highlighting the 4 characteristics (
positive remodeling, napkin ring lesions, spotty calcification, and low attenuation non-calcified plaque)

Author Response

Alyami et al. present a review article about the assessment of the prevalence of non-calcified plaque on computed tomography angiography in symptomatic and asymptomatic individuals. Furthermore, they assessed the prognostic value of NCP. Although the topic is interesting and the manuscript well-written, some considerations need to be clarified.

  1. Introduction: the authors cite the ACC/AHA guidelines and SCCT guidelines. Please, also cite the point of view of ESC Guidelines (Knuuti J, et al. ESC Scientific Document Group. 2019 ESC Guidelines for the diagnosis and management of chronic coronary syndromes. Eur Heart J. 2020 Jan 14;41(3):407-477. doi: 10.1093/eurheartj/ehz425).

Thank you for your valauble comment , we have cited the above refernace in our study

  1. Discussion: Please, suggest the therapeutic strategy that can be useful for the plaque regression based on the current literature.

Thank you and we have added a study that discussed this aspect as following: According to a meta-analysis published by Linn et al. aimed to evaluate the effect of statin treatment on total plaque volume and mean volume regression. The results showed that the group on intensive statin therapy experienced a higher reduction in total plaque volume and mean volume regression compared to the moderate-intensity statin group. In contrast, the control group exhibited significant increases in total plaque volume and progression of the mean volume.

  1. Please, insert a new paragraph “gaps in evidence and future directions” after the discussion.
  2. Thank you for your comment. We have thoroughly revised and editted our discussion to ensure that it includes the mentioned important information.

The authors satisfactorily answered their main question to evaluate the prevalence of non-calcified plaque on computed tomography angiography in symptomatic and asymptomatic individuals and its prognostic value. The topic is original and adds interesting information on importance of detecting the non-calcified plaque during early stages to reduce the future incidence of adverse cardiovascular events. The conclusions are consistent with the current evidence but the references should be improved.  A figure about high-risk plaque on computed tomography angiography could be added by highlighting the 4 characteristics (positive remodeling, napkin ring lesions, spotty calcification, and low attenuation non-calcified plaque).

Thank you for your insightful feedback on our manuscript. We are happy to hear that you found our study's question adequately addressed and that the topic is considered original and significant in shedding light on the importance of detecting non-calcified plaque in early stages for reducing adverse cardiovascular events. Regarding the references, we appreciate your suggestion, and we have taken it into consideration and added new refernces.  Furthermore, we have incorporated your suggestion of adding a figure to highlight high-risk plaque characteristics on computed tomography angiography. The figure now includes the four key characteristics: positive remodeling, napkin ring lesions, spotty calcification, and low attenuation non-calcified plaque to enhance the visual representation of our paper. Once again, thank you for your valuable input, as it has helped us enhance the quality and impact of our manuscript. We hope the revised version meets your expectations and addresses all the aspects you highlighted.

Round 2

Reviewer 1 Report

Authors addressed all my concerns.

Fine

Author Response

Dear Editors, please find our response to the journal’s inquire for our submitted review article: non-calcified coronary artery plaque on Coronary CT Angiogram: prevalence and Significance

Reviewer one comment: Authors addressed all my concerns.

Reviewer two comment: The authors have responded satisfactorily to my requests and comments.

Academic Editor comment:

In summary this is an interesting paper which merits publication, provided two faults are corrected diligently.

Please rewrite results section and table, the last two columns. 

Thanks for your comment, we have reviewed them and applies some change please specify more if any further change needed.   

I missed a short paragraph on how the papers were selected for their review. Which databanks? Which key words? Inclusion/exclusion criteria?

The selection of studies for this systematic review on the prevalence and significance of non-calcified coronary artery plaques on Coronary CT Angiograms (CTA) was as following:   Initial searches were conducted on the PubMed database, utilizing a combination of relevant Medical Subject Headings (MeSH) terms and keywords such as "non-calcified coronary artery plaque," "coronary CT angiogram," "prevalence," and "significance." The search aimed to identify studies published in the English language without imposing limitations on publication dates.

Inclusion criteria were defined to encompass studies specifically investigating non-calcified coronary artery plaques as detected by CTA and providing insights into both their prevalence and clinical relevance. Studies falling under the categories of reviews, case reports, conference abstracts, and those lacking essential data were excluded. To uphold the credibility of the review, only peer-reviewed, full-text articles were incorporated.

Other journal comments:

we find that Figure 1 and 2 has citation, please make sure that permission has been obtained and there is no copyright issue.

Permission has been obtained and uploaded a copy to this email for your references, please let us know if you need any further information.

We find that there are still some paragraphs are similar to the previous study, we have highlighted those parts in the attachment. To avoid potential conflict, please revise the manuscript either by changing the way of 
expression or the structure of sentence.

thanks for your valuable feedback, we have rewritten all of them again. 

Reviewer 2 Report

The authors have responded satisfactorily to my requests and comments.

Author Response

(The authors gave the same response as above.)
